# Comparative Study on Chloride Binding Capacity of Cement-Fly Ash System and Cement-Ground Granulated Blast Furnace Slag System with Diethanol-Isopropanolamine

**DOI:** 10.3390/ma13184103

**Published:** 2020-09-16

**Authors:** Huaqing Liu, Yan Zhang, Jialong Liu, Zixia Feng, Sen Kong

**Affiliations:** China Electric Power Research Institute, Beijing 100192, China; lhq@epri.sgcc.com.cn (H.L.); liujialong@epri.sgcc.com.cn (J.L.); fengzx@epri.sgcc.com.cn (Z.F.); kongsen@epri.sgcc.com.cn (S.K.)

**Keywords:** chloride binding capacity, fly ash, ground granulated blast furnace slag, diethanol-isopropanolamine

## Abstract

Steel bar corrosion caused by chloride was one of the main forms of concrete deterioration. The promotion of chloride binding capacity of cementitious materials would hinder the chloride transport to the surface of steel bar, thereby alleviating the corrosion and mitigating the deterioration. A comparative study on binding capacity of chloride in cement-fly ash system (C-FA) and cement-ground granulated blast furnace slag system (C-GGBS) with diethanol-isopropanolamine (DEIPA) was investigated in this study. Chloride ions was introduced by adding NaCl in paste, and the chloride binding capacity of the paste samples at 7 d and 60 d was examined. The hydration process was discussed via the testing of hydration heat and compressive strength. The hydrates in hardened paste was characterized by X-ray Diffractometry (XRD), Thermo Gravimetric Analysis (TGA), and Scanning Electron Microscope (SEM). The effect of DEIPA on dissolution of aluminate phase and compressive strength was discussed as well. These results showed that DEIPA could facilitate the hydration of C-FA and C-GGBS system, and the promotion effect was higher in C-FA than that in C-GGBS. DEIPA also increased the binding capacity of chloride in C-FA and C-GGBS systems. One reason was the increased chemical binding, because DEIPA facilitated the dissolution of aluminate to benefit the formation of Friedel’s salt. Other reasons were the increased physical binding and migration resistance. By contrast, DEIPA presented greater ability to increase chloride binding capacity in C-FA system, because DEIPA showed stronger ability to expedite the dissolution of aluminate of FA than that of GGBS, which benefited the formation of FS, thereby promoting the chemical binding. Such results would give deep insight into using DEIPA as an additive in cement-based materials.

## 1. Introduction

In sea construction engineering, the utilization of sea resources, such as sea sands, coral aggregates, and sea water, has attracted increasing attention, not only because of the low cost, but also due to the shortage of resources from the mainland [1]. However, these sea resources brought harmful ions into the cementitious system, such as sulfate and chloride [2,3,4]. The negative effect of these ions on reinforced concrete attracted great attention; among these negative effects, the steel bar corrosion resulting from chloride ions was accepted as the most important one, because once the corrosion happened, the mechanical performance of concrete would be obviously deteriorated and even invalided with time going on.

In the literature, it was reported that the harmful ions could be immobilized in material systems [5,6,7]. The chloride ions in hardened cement paste system were divided into three types. One was the chemical bound chloride. This kind of chloride appeared in Friedel’s salt (3CaO·Al_2_O_3_·CaCl_2_·10H_2_O, FS) and Kuzel’s salt (3CaO·Al_2_O_3_·0.5CaCl_2_·0.5CaSO_4_·10H_2_O, KS) [8,9,10]. The second one was the physical adsorbed chloride, which existed in layer structure of C-S-H gel and, generally, greater amount of C-S-H gel could adsorb more amount of chloride ions [11]. The third one was the free chloride ions, which could freely migrate in the system, with great potential risk to steel bar corrosion. It was evidenced that the chemically bound chloride presented little negative effect on steel bar corrosion. Therefore, three measures could be taken in order to promote the binding capacity of chloride in cementitious system. One was to promote the formation of FS or KS to chemically bind chloride ions; the second was to hasten the hydration to produce more amount of C-S-H gel to physically adsorb the chloride ions; the last was to improve the pore structure to hinder the migration of chloride ions. For example, one reason for supplementary cementitious materials (SCM), such fly ash (FA) and ground granulated blast furnace slag (GGBS), to promote the binding capacity was that the dissolution of aluminates could be facilitated in the process of pozzolanic reaction, which could expedite the formation of FS or KS, thereby promoting chemical binding [12,13]. The other reason was that pore structure and the hydration degree were also improved, which promoted the migration resistance and physical binding. Additionally, SCM was commonly used in cement-based materials to improve the basic performance, such as volume stability, hydration process, and durability [14,15,16,17], and it also showed great environmental and economic benefits [18,19,20]. The hydration process of SCM was closely related to chloride binding [21,22,23]. Furthermore, the addition of alkanolamine could hasten the hydration of SCM to increase the chloride binding capacity [24]. For example, triisopropanolamine could promote the hydration of FA and GGBS, and the binding capacity of chloride in system was also promoted, because triisopropanolamine facilitated the dissolution of aluminate in FA and GGBS in order to hasten the formation of KS or FS from AFm. It was also reported that triethanolamine (TEA) could expedite the dissolution of FA, and the dissolution of aluminate phase was greatly promoted [25]. Obviously, alkanolamine could increase chloride binding capacity by promoting the dissolution of aluminate phase.

In the literature, diethanol-isopropanolamine (DEIPA) was reported to promote the hydration of cementitious system [26,27], and it was widely used as grinding aid agent or accelerator in Portland cement system. However, it seemed that little attention was paid to chloride binding that was influenced by DEIPA. In this study, comparative study on the binding capacity of chloride in cement-fly ash system and cement-ground granulated blast furnace slag system with the addition of DEIPA was investigated. The mechanism behind was also discussed in terms of hydration process and hydrates in hardened paste. Such results expected to offer experience of the enhancement of chloride binding capacity in cementitious materials. Such results would give deep insight into using DEIPA as an additive in cement-based materials.

## 2. Materials and Test Methods

### 2.1. Materials

#### 2.1.1. Cement

An ordinary Portland cement (42.5 grade), coal-fired FA (grade II) obtained from coal-fired plant, GGBS (S95 grade), were used in this study. The particle sized distribution, which was obtained by laser particle size analyzer (Mastersizer 2000, Malvin Inc. Malvern, London, UK), is shown in Figure 1. Cement, FA, and GGBS was characterized by X-ray Fluorescence (XRF, PANalytical.B.V, Almelo, Netherlands), and the results are shown in Table 1.

#### 2.1.2. Chemicals

Diethanol-isopropanolamine (DEIPA, reagent-grade, ≥96.0% purity) was used in this study, and the molecular structure was shown in Figure 2. It was noted that the dosage of DEIPA that was used in the experiments was marked as the solid amount.

#### 2.1.3. Preparation of the Samples

NaCl was used to introduce chloride into the system. Two kinds of binders were designed. cement-ground granulated blast furnace slag binder (C-GGBS) was composed of 30 wt.% GGBS and 70 wt.% cement, and cement-fly ash binder (C-FA) was composed of 30 wt.% FA and 70 wt.% cement. the mix proportions were shown in Table 2. The water/binder ratio was 0.38 by weight. DEIPA (0, 0.03%, 0.05%, and 0.10% of binder by weight) and NaCl (reagent-grade, 1.11 wt.% of binder by weight) were mixed with water in advance. The pastes were prepared in accordance with the Chinese standard GB/T 8077-2012. 40 mm × 40 mm × 40 mm moulds were used to cast the paste samples. These samples were then cured in a chamber ((20 ± 1) °C, ≥90% R.H.) for 1 d, followed by demoulding process. After that, the samples were further cured to 7 d and 60 d age.

The hardened samples were crushed into small pieces. These pieces were soaked in ethanol for 48 h in order to terminate hydration. Thereafter, a vacuum with 25 ± 1 °C was used for drying 1 d. Some small pieces were selected for the characterizations of pore structure and morphology by Mercury intrusion porosimetry (MIP, Poremaster GT-60, Kangta Instrument Company, Beijing, China) and Scanning Electron Microscope (SEM). Pieces with particle sized less than 0.15 mm were prepared for the measurement of chloride binding ratio (CBR). Other were ground into powders to pass through a 45 μm sieve and used for the measurement of X-ray Diffractometry (XRD, D/Max-RB, Cu-Kα (1.541874 Å), Bruker, Beijing, China) and Thermo Gravimetric Analysis (TGA, German-resistant STA449F3, NETZSCH, Selb, Germany).

### 2.2. Test Methods

#### 2.2.1. Chloride Binding Ratio

Chloride binding ratio was evaluated on the basis of the Chinese standard SL 352-2006 (Chinese Hydraulic Concrete Test standards, Ministry of Water Resources of the People’s Republic of China, Beijing, China).

The suspension was prepared from 10.0 g paste powder and 100 mL deionized water, and it was severely shocked for 2 min. Subsequently, it was stored for 24 h. The upper supernatant was obtained by filtration. Phenolphthalein was added as an indicator, and it was neutralized to be pH = 7 by dilute sulfuric acid. Mohr method was used to test free chloride ion, and the chemical reactions happened as Equations (1) and (2). To be more precise, 20.0 mL filtrate was titrated with 0.02 mol/L AgNO_3_ while using K_2_CrO_4_ solution (5 wt%, 0.5 g) as an indicator. When the brick red precipitate Ag_2_CrO_4_ appeared, the end point of the titration happened. Free chloride ion content was calculated as Equation (3). Each filtrate was tested three times, and the average was the result.

The initial total chloride content in the sample can be calculated based on the added amount of NaCl. During alcohol immersion and drying, 25% free water was lost based on the laboratory test. Therefore, the total chloride content (C_t_) is 5.405 mg·g^−1^ in dried hardened cement paste. The difference between C_t_ and C_f_ is the content of bound chloride. Additionally, the chloride binding ratio was calculated as Equation (4) [28,29,30].
AgNO_3_+Cl^−^→AgCl↓+NO_3_^−^(1)
2AgNO_3_+K_2_CrO_4_→Ag_2_CrO_4_↓+2KNO_3_(2)
(3)$$C_{f}=\frac{CV_{3}\times 0.03545}{G\times \frac{V_{2}}{V_{1}}}\times 100\%$$
(4)$$CBR=\frac{C_{t}{-C}_{f}}{C_{t}}\times 100\%$$

Wherein, *C_f_*: free chloride ion content in the paste, %; C: AgNO_3_ concentration, mol/L; *G*: sample mass, g; *V*_1_: volume of water soaking sample, mL; *V*_2_: the volume of the filtrate used for titration, mL; *V*_3_: volume of the consumption of silver nitrate solution, mL; and, *C_t_*: one is the total chlorine ion content (*C_t_*), which was dependent on amount of NaCl added in the sample preparation.

#### 2.2.2. Compressive Strength

According to Chinese standard GB/T 17669.3-1999, the compressive strength was tested, with the loading speed of 2.4 kN/s. In each group, three specimens were considered and the average was calculated as a final result.

#### 2.2.3. Hydration Heat

DEIPA (0, 0.03%, and 0.05% of binder) was mixed with water in advance. C-FA (70% cement and 30% FA) binder and C-GGBS (70% cement and 30% FA) binder for the measurement were prepared with a water/binder weight ratio of 0.38. The hydration heat was then examined with an isothermal calorimetry (TAM AIR C80, SETARAM, Shanghai, China).

#### 2.2.4. Pore Structure

The pore structure was characterized by mercury intrusion porosimetry (MIP, Poremaster GT-60, Kangta Instrument Company, Beijing, China).

In the process of measurement, the surface energy increased by pores and external pressure on mercury could be expressed in Equation (5):(5)$$dW=-PdV=-\gamma_{L}cos\theta dS$$
where *W* denotes surface energy of the pore; *V* is the volume of the intruded mercury; *P* represents the pressure of mercury; *γ*_L_ means surface tension of mercury; *θ* is the contact angle of mercury on pore surface; and, *S* is the pore surface area.

It was reported by Zhang and Li that the accumulated intrusion surface energy (*W_n_*) and the accumulated mercury intrusion surface (*Q_n_*) could be expressed as Equation (6) [31]. *W_n_* and *Q_n_* could be expressed as Equations (7) and (8).
(6)$$LogW_{n}=LogQ_{n}+C$$
(7)$$W_{n}=\overset{n}{\underset{i=1}{\sum }}P_{i}\Delta V_{i}$$
(8)$$Q_{n}=r_{n}^{2-D}V_{n}^{3/D}$$
where, *C* is a constant; *V_n_* is the accumulated volume of mercury; *r_n_* is the smallest pore radius; *P_i_* and *V_i_* denote for the pressure of mercury and the volume of the intruded mercury at step i; and, *D* stands for the fractal dimension of pore surface [32,33,34].

Accordingly, Equation (6) can be expressed as Equation (9).
(9)$$log\left(\frac{W_{n}}{r_{n}^{2}}\right)=D\;log\left(\frac{V_{n}^{3/D}}{r_{n}}\right)+C$$

The fractal dimension of the pore surface was obtained by the slope of the curve log (*V_n_*^1/3^/*r_n_*) versus log (*W_n_*/*r_n_*^2^), with larger than 0.99 of correlation coefficient.

#### 2.2.5. Phase Analysis

SEM: the tiny piece samples were prepared for SEM measurement, which was conducted by Field Emission Scanning Electron Microscope (FE-SEM, QUANTA FEG 450, FEI Co, Hillsboro, OR, USA).

XRD: the hydration products were examined by X-ray diffraction (D/Max-RB, Cu-Kα (1.541874 Å), Bruker, Beijing, China), and 5–70° range was scanned.

TGA: the powder samples of cement-GGBS paste were examined with the comprehensive thermal analyzer (German-resistant STA449F3, NETZSCH, Selb, Germany), and the room temperature ranged from the room temperature to 1000 °C at a temperature rise rate of 10 °C/min. The decomposition of calcium hydroxide (CH) occurred at 400–500 °C (Equation (10)) [35,36]. The amount of CH in hydrate was calculated from Equation (11):(10)$${\mathrm{Ca}(\mathrm{OH})}_{2}\to {\mathrm{CaO}+H}_{2}O$$
(11)$$M_{\mathrm{Ca}{\left(\mathrm{OH}\right)}_{2}}=\frac{74}{18}{\;M}_{H_{2}O}$$

$M_{\mathrm{Ca}{\left(\mathrm{OH}\right)}_{2}}$: mass of CH; ${\;M}_{H_{2}O}\;$: the loss weight at 400–500 °C.

## 3. Results and Discussion

### 3.1. Effect of DEIPA on Chloride Binding Capacity

Figure 3 shows the chloride binding ratio (CBR) of C-FA and C-GGBS system with various dosage of DEIPA at the age of 7 d and 60 d. From the Figure 3a, it was found that, in the C-FA system, 0.03% DEIPA increased the CBR value from 38% to 43%, with an increase by 13%, and that in C-GGBS was by 5%. 0.10% DEIPA increased the CBR to 47.5% in C-FA system and 47.6% in C-GGBS system, with an increase by 23% and 16%. This indicated that DEIPA increased the CBR value of both C-FA and C-GGBS systems at the age of 7 d, and more dosage resulted in higher increase ratio. By contrast, DEIPA seemed more efficient in C-FA system at the age of 7 d. From the Figure 3b, it was observed that at the age of 60 d, the CBR value of C-FA system was higher than that of C-GGBS, regardless of whether DEIPA was added or not. This indicated that C-FA system presented stronger chloride binding capacity than that of C-GGBS at the age of 60 d. Furthermore, it was found that the increase in dosage of DEIPA increased the CBR value of both C-FA and C-GGBS systems. This demonstrated that DEIPA could increase the chloride binding capacity of the C-FA and C-GGBS systems. By contrast, the increase ratio in the C-FA system was higher than that in C-GGBS system. With 0.03% DEIPA, the increase ratio in CBR of C-FA was 9%, while that in C-GGBS was 5%; with 0.06% DEIPA, the increase ratio was 18% in C-FA and 15% in C-GGBS; with 0.10%, it was 27% in C-FA and 120% in C-GGBS. These results illustrated that DEIPA exhibited higher capacity to increase chloride binding capacity in the C-FA system than that in the C-GGBS system.

It was reported that the chloride binding capacity of cement-based materials depended on three aspects: the first one is the chemical binding, and it was decided by the amount of FS and KS in hydrates. The second one was physical binding, and it was dependent on amount of hydrates, especially C-S-H gel, because these hydrates could physically adsorb the chloride ion. The third one was related to the pore structure, because the refined pore structure with complex channels could block the transport of chloride ions. These three aspects were explained in the following text.

### 3.2. Hydration Process

#### 3.2.1. Hydration Heat

Figure 4 shows the hydration heat of C-FA and C-GGBS with various dosages of DEIPA. In the literature, three peaks were observed in C-FA system and C-GGBS system. The first peak was located at the first few minutes, which was attributed to the initial dissolution of ions and the hydration of tricalcium aluminate. The second peak appeared at around 20 h in C-FA, and this was due to the hydration of tricalcium silicate and formation of the C-S-H gel. The third peak was attributed to the transformation of ettringite to AFm. From Figure 4a, in C-FA system, two peaks were clearly observed, at first few minutes and at around 20 h. The third peak appeared at about 30 h. With the addition of DEIPA, the first and second peaks seemed no obvious change, but the third peak was greatly promoted, and greater dosage resulted in more promotion. This result indicated that DEIPA facilitated the transformation of ettringite to AFm. In C-GGBS system, the same tendency was found. From Figure 4b, it was observed at DEIPA promoted the release of the hydration heat in both C-FA and C-GGBS systems. By contrast, the promotion seemed more efficient in the C-GGBS system than that in the C-FA system. As shown in Table 3, without DEIPA, the hydration heat of C-FA system was 8.20 J/g, and that in C-GGBS system was 23 J/g at the age of 6 h. The same tendency was found at the following age. This was because the hydration activity of FA was much lower than that of GGBS at the early age. Furthermore, it was also found that, in both C-FA and C-GGBS systems, DEIPA promoted the release of the hydration heat, indicating that the addition of DEIPA could facilitate the hydration of both C-FA and C-GGBS systems. By contrast, with addition of DEIPA, the hydration heat of C-FA was also lower than that of C-GGBS within 7 d age.

#### 3.2.2. Compressive Strength

Figure 5 shows the effect of DEIPA on the compressive strength of C-FA and C-GGBS systems. In Figure 5a, it was found that, at 7 d, DEIPA increased the compressive strength of both C-FA and C-GGBS, and greater dosage resulted in a higher increase. By contrast, the increase ratio was much higher in C-GGBS than that in C-FA. From Figure 5b, DEIPA also increased the 60-d compressive strength in both C-FA and C-GGBS systems, and more dosage also led to stronger effect. By contrast, the increase ratio was much higher in the C-FA system than that in C-GGBS, which was opposite to the results at the age of 7 d. It was proved in the literature that the increased strength was closely related to the hydration. The increased strength of both C-FA and C-GGBS by DEIPA indicated that DEIPA could greatly facilitate the hydration of the systems, not only at the age of 7 d, but also 60 d. By contrast, at the age of 7 d, the facilitation seemed much stronger in C-GGBS system, while at the age of 60 d, the opposite was true.

### 3.3. Hydrates Analysis

#### 3.3.1. XRD

Figure 6 shows the XRD patterns of C-FA and C-GGBS samples hydrated for 7 d and 60 d. From Figure 6a, Friedel’s salt (3CaO·Al_2_O_3_·CaCl_2_·10H_2_O, PDF# 42-0558) and calcium hydroxide (CH, Ca(OH)_2_, PDF# 04-0733) were clearly observed in the C-FA and C-GGBS samples. The presence of CH peak indicated the cement hydration. It seemed that no obvious difference was observed in peak intensity of CH between the samples with and without DEIPA, and this result indicated that the addition of DEIPA exhibited no obvious effect on formation of CH. The peak intensity of FS was increased by adding DEIPA, not only in C-FA system but also in C-GGBS system, and this result indicated that DEIPA facilitated the formation of FS in both C-FA and C-GGBS systems. From Figure 6b, the same results were observed; in comparison with the samples without DEIPA, the peak intensity of FS in both C-FA and C-GGBS systems was increased by adding DEIPA, and this indicated that the formation of FS was also facilitated by adding DEIPA at the age of 60 d.

#### 3.3.2. TG

In the literature, it was reported that the peaks at 50–200 °C were because of the dehydration of C-S-H gel, AFt, and AFm phase; that at 280–390 °C was mainly attributed to the decomposition of FS; the peaks around 400–500 °C mainly result from the dehydration of calcium hydroxide, which around 550–800 °C was due to the decomposition of calcium carbonate, resulting from the carbonation in the process of sample preparation [36,37]. To further analyze the hydrates, the samples were tested by TG, and the weight loss at 50–200 °C, 280–390 °C, and 400–500 °C were calculated; the results are shown in Table 4 and Table 5. As shown in Table 4, it was found that with 0.06% DEIPA, the weight loss was increased at 50–200 °C, 280–390 °C, and 400–500 °C, and this result indicated that DEIPA facilitated the hydration of both C-FA and C-GGBS systems at the age of 7 d. It was noted that the weight loss at 50–200 °C and 280–390 °C in C-GGBS sample was higher than that of C-FA, and this indicated that, at the age of 7 d, greater amount of C-S-H gel and FS in hydrates was produced. It was due to the higher hydration activity of GGBS than FA at the early age. The content of CH in C-GGBS was lower than that of C-FA, and this was also because GGBS with higher hydration activity at the early age presented stronger ability to consume CH [38,39].

As shown in Table 5, it was also observed that the weight loss at 50–200 °C, 280–390 °C, and 400–500 °C was increased by DEIPA. This result indicated that DEIPA facilitated the hydration of both C-FA and C-GGBS systems at the age of 60 d. It also demonstrated that DEIPA in both C-FA and C-GGBS systems could facilitate the formation of FS. By contrast, at 50–200 °C, weight loss of C-FA samples was lower than that of C-GGBS, because of the lower activity of FA than GGBS. At 280–390 °C, weight loss of C-FA samples was higher than that of C-GGBS, and this indicated that, at the age of 60 d, the amount of FS in C-FA system was higher than that in C-GBBS. 0.06% DEIPA increased the weight loss of 50–200 °C and 280–390 °C in both C-FA and C-GGBS systems, and this result indicated that DEIPA facilitated the hydration and the formation of FS and C-S-H gel.

#### 3.3.3. SEM

The hydrates were investigated by SEM, and the results are shown in Figure 7 and Figure 8. As shown in Figure 7a,b, FS with sheet-typed structure was clearly found, and this indicated that, at the age of 7 d, FS was produced, which was one of the reason for the chloride binding of the system. By contrast, DEIPA seemed to expedite the formation of FS. At the age of 60 d, C-S-H gel in both Figure 7c,d was observed, and this was because of the continuous hydration of the system. By contrast, DEIPA seemed to densify the hardened paste, and the possible reason was that DEIPA could facilitate the hydration of the system.

From Figure 8a, sheet-typed FS was also found, while this structure was also observed in Figure 8b. The result indicated that FS was produced in C-FA system at the age of 7 d. From Figure 8b, a little amount of sheet-typed hydrates was observed on the surface of FA. This implied that, with the addition of DEIPA, a little hydration of FA took place at the age of 7 d. Furthermore, from Figure 8c,d, hydrates were observed on the FA surface, and this indicated that the pozzolanic reaction of FA obviously happened at the age of 60 d. Sheet-typed hydrates were also found, which indicated the formation of FS at 60 d, which agreed with the results of XRD. By contrast, DEIPA seemed to expedite the hydration of FA and the formation of the FS at 60 d.

Based on discussion mentioned above, it could be concluded that DEIPA facilitated the hydration of both C-FA and C-GGBS systems, and the formation of FS was also facilitated. By contrast, at the early age, the facilitation of DEIPA seemed to be higher in the C-GGBS system and, at the later age, it was higher in the C-FA system.

### 3.4. Pore Structure

Pore structure was reported to influence the ion transport, volume stability, and mechanical performance [40,41,42,43,44]. The pore structure of C-FA and C-GGBS system was evaluated by MIP, and the pore distribution is depicted in Figure 9. From Figure 9a, in C-FA system, DEIPA increased the most probable aperture (MPA) at 7 d, and the same result was also found in Figure 9b. In C-GGBS system, this tendency seemed no appearance.

The porosity was calculated, and the result is shown in Table 6. It was found that 0.06% DEIPA increased the 7-d porosity of C-FA system from 0.1207 mL/g to 0.1336 mL/g. At the age of 60 d, the increase was from 0.0888 mL/g to 0.0902 mL/g. These results indicated that DEIPA could increase the porosity of C-FA system, and this agreed with the result in the literatures [45].

Based on the MIP data, the fractal characteristics were calculated, and the results are shown in Figure 10 and Figure 11. The microporous structure could also be reflected by fractal dimension which was related to the complexity of pore structure. As reported in the literature, the packing patterns of the hydrated binder particles could be reflected by the Ds-a of the macro fractal region, and the microstructure of the C-S-H gel could be denoted by the Ds-i of the differential region [32,33]. It was observed from Figure 10 that in C-FA system at the age of 7 d, DEIPA increased Ds-a and Ds-i from 2.37 to 2.46 and from 2.69 to 2.76. The increases were also found in C-GGBS system. This result demonstrated that the addition of DEIPA in C-FA and C-GGBS disordered the transport tunnel, which complexed the pore structure at 7 d, which would hinder the transport of chloride ions. From Figure 11, the same results were also found, which indicated that DEIPA also disordered the transport tunnel and complexed the pore structure at 60 d. This indicated that the addition of DEIPA in both C-FA and C-GGBS systems could increase the complexity of pore structure.

Based on discussion about the pore structure, it was concluded that DEIPA could increase the complexity of the pore structure of both C-FA and C-GGBS, and this benefited in hindering the transport of chloride ions.

### 3.5. Discussion

#### 3.5.1. Effect of DEIPA on Hydration Process

Based on the experiment results, it was found that DEIPA could facilitate the hydration of the C-GGBS and C-FA systems. One reason was that DEIPA could hasten the hydration of cement, because DEIPA could complex with ferric phase to hinder its precipitation so that the dissolution of phases could easily take place [46,47,48]. The other reason was that DEIPA could facilitate the dissolution of FA and GGBS, and this could accelerate the pozzolanic reaction. However, effect of DEIPA on dissolution of FA and GGBS presented an obvious difference.

As shown in Table 7, it was seen that the age of 7 d, 20 g/L DEIPA increased aluminate concentration in FA suspension from 150 mg/L to 430 mg/L, with an increase by 187%; that for 60 d was from 400 mg/L to 1400 mg/L, with an increase by 367%. In GGBS suspension, which for 7 d was from 80 mg/L to 230 mg/L, with an increase by 92%, and that for 60 d was increase by 100%. This result indicated that 20 g/L DEIPA increased the dissolution of Al in FA and GGBS at the age of 7 d and 60 d. These aluminates could take part in hydration, and the hydration of FA and GGBS was expedited. By contrast, the promotion of Al dissolution by DEIPA seemed to be much higher in C-FA system than that in C-GGBS system.

#### 3.5.2. Effect of DEIPA on Chloride Binding Capacity

Chloride binding capacity in cementitious materials was related to chemical binding, physical binding, and migration resistance. Firstly, chemical binding was dependent on the formation of FS or KS in system. A greater amount of aluminate in system to take part in hydration could expedite the formation of FS and KS. In the presence of DEIPA, the dissolution of aluminate in FA and GGBS could be facilitated, and this could facilitate the aluminate to take part in hydration reaction. Accordingly, the formation of FS could be expedited, and this was also proved by XRD. Accordingly, DEIPA could increase the chemical binding of the C-FA and C-GGBS system. By contrast, as DEIPA showed greater ability to facilitate the dissolution of aluminate in C-FA than that in C-GGBS, the chemical binding capacity that was increased by DEIPA should be higher in the C-FA system. Furthermore, the hydration of C-FA and C-GGBS was facilitated by adding DEIPA, and this indicated that C-S-H gel formed in system was facilitated. As the physical binding of chloride depended on the amount of C-S-H gel in hydrates, DEIPA also increased the physical binding of C-FA and C-GGBS system. By contrast, at the early age, even though DEIPA showed higher ability to facilitate the hydration of C-FA system than that of C-GGBS system, the hydration activity of GGBS at the early age was much higher than that of FA. Accordingly, at the early age, the C-GGBS system presented higher physically binding capacity than that of C-FA system. Additionally, the ion transport and migration were dependent on pore structure. It was evidenced that DEIPA could increase the complex of the pore structure in both C-FA and C-GGBS systems, and this demonstrated that DEIPA could increase the migration resistance of these two systems. As a result, DEIPA added in C-FA and C-GGBS systems could increase the migration resistance, physical binding, and chemical binding of these two systems. At the early age, the C-GGBS system presented higher chloride binding capacity, and at the later time, because of the dissolution of aluminate phase in FA, higher binding capacity in C-FA system was observed. By contrast, DEIPA showed stronger promotion of binding capacity of chloride in C-FA than that in C-GGBS, especially at the later age.

## 4. Conclusions

The effect of DEIPA on chloride binding capacity of cement-fly ash system and cement-ground granulated blast furnace slag system was systematically discussed, and the conclusion was drawn as follow:(1)DEIPA facilitated the hydration and increased the compressive strength of both C-FA and C-GGBS systems, and the main reason was that DEIPA facilitate the pozzolanic reaction of FA and GGBS.(2)DEIPA increased the chloride binding capacity of both the C-FA and C-GGBS systems. One reason was that DEIPA facilitated the dissolution of aluminate to hasten the formation of FS, with contribution to the chemical binding. Other reasons were that DEIPA increased the complex of pore structure, which increased the migration resistance and expedited the hydration of the system to produce more amount of C-S-H gel which benefited to the physical binding.(3)By contrast, DEIPA presented greater ability to elevate the chloride binding capacity in the C-FA system. The reason for this was that DEIPA showed stronger ability to expedite the dissolution of aluminate of FA than that of GGBS.

## Figures and Tables

**Figure 1 materials-13-04103-f001:**
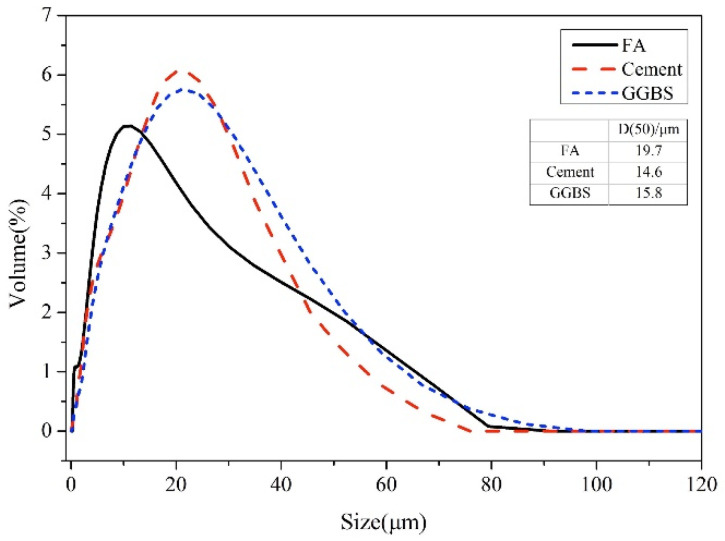
Particle size distribution of fly ash (FA), ground granulated blast furnace slag (GGBS), and cement.

**Figure 2 materials-13-04103-f002:**
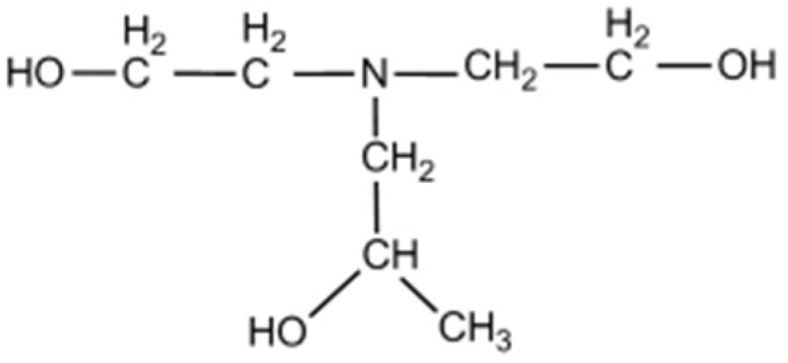
Molecular structure of diethanol-isopropanolamine (DEIPA).

**Figure 3 materials-13-04103-f003:**
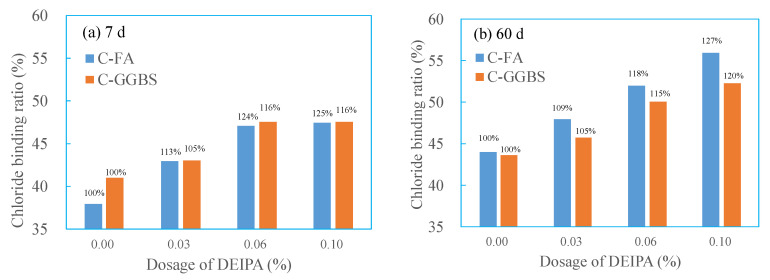
Effect of chloride binding capacity of cement-fly ash system (C-FA) and cement-ground granulated blast furnace slag (C-GGBS), (**a**) 7 d, (**b**) 60 d.

**Figure 4 materials-13-04103-f004:**
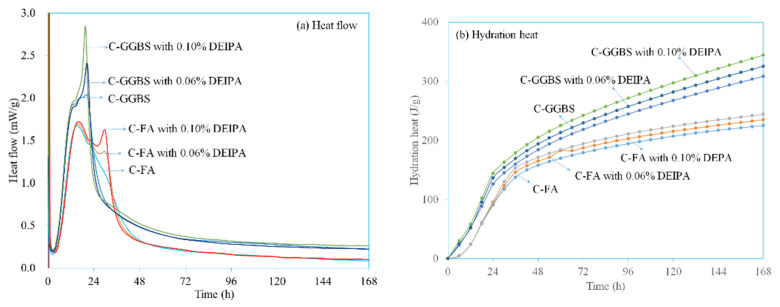
Hydration heat of the paste with DEIPA, (**a**) heat flow, (**b**) hydration heat.

**Figure 5 materials-13-04103-f005:**
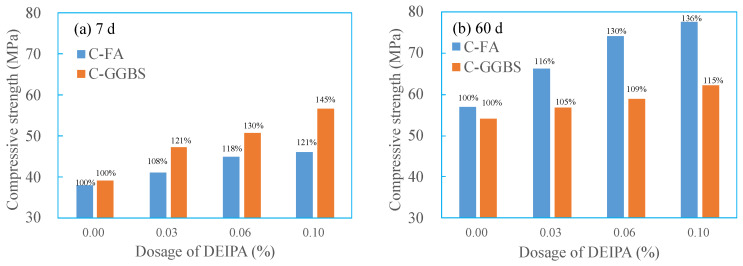
Effect of DEIPA on compressive strength of the paste, (**a**) 7 d, (**b**) 60 d.

**Figure 6 materials-13-04103-f006:**
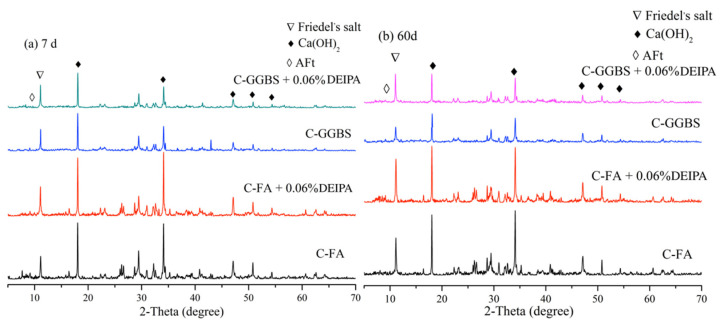
X-ray Diffractometry (XRD) patterns of samples hydrated for 7 d and 60 d, (**a**) 7 d, (**b**) 60 d.

**Figure 7 materials-13-04103-f007:**
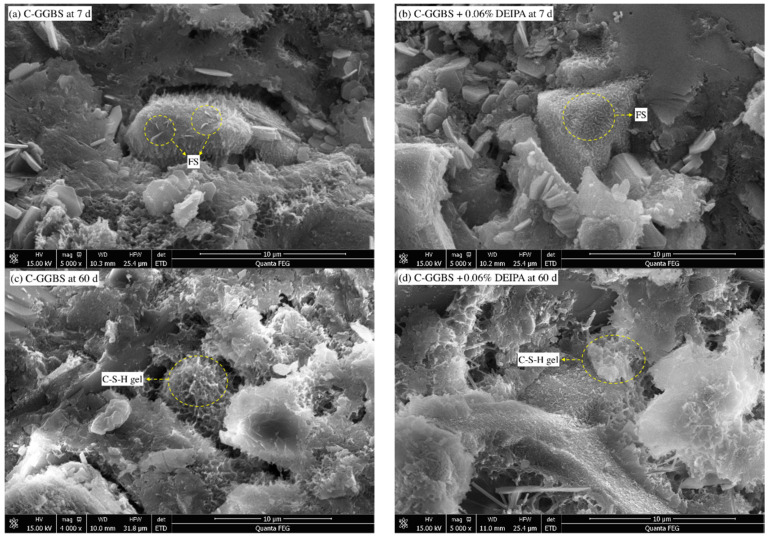
Scanning Electron Microscope (SEM) images of C-GGBS, (**a**) C-GGBS at 7 d, (**b**) C-GGBS with 0.06% DEIPA at 7 d, (**c**) C-GGBS at 60 d, (**d**) C-GGBS with 0.06% DEIPA at 60 d.

**Figure 8 materials-13-04103-f008:**
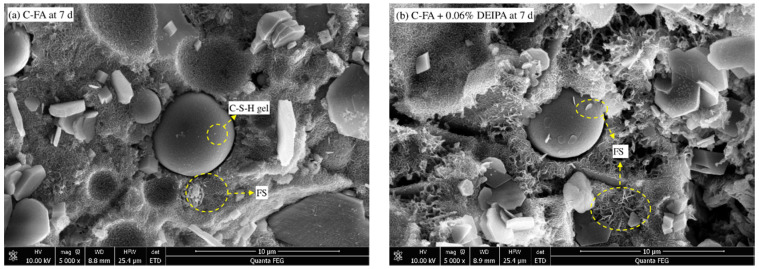
SEM images of C-FA, (**a**) C-FA at 7 d, (**b**) C-FA with 0.06% DEIPAat 7 d, (**c**) C-FA at 60 d, (**d**) C-FA with 0.06% DEIPA at 60 d.

**Figure 9 materials-13-04103-f009:**
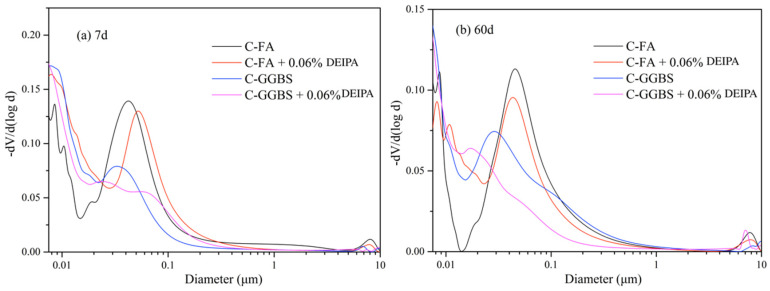
Pore distribution of C-FA and C-GGBS systems, (**a**) 7 d, (**b**) 60 d.

**Figure 10 materials-13-04103-f010:**
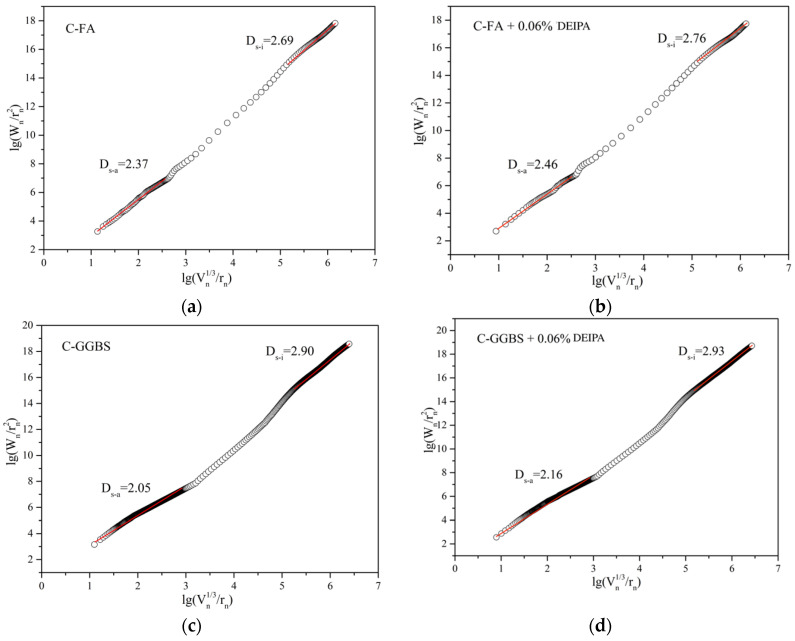
Logarithm plots of W_n_/r_n_^2^ versus V_n_^1/3^/r_n_ of C-FA/GGBS systems at 7 d, (**a**) C-FA, (**b**) C-FA with 0.06% DEIPA, (**c**) C-GGBS, (**d**) C-GGBS with 0.06% DEIPA.

**Figure 11 materials-13-04103-f011:**
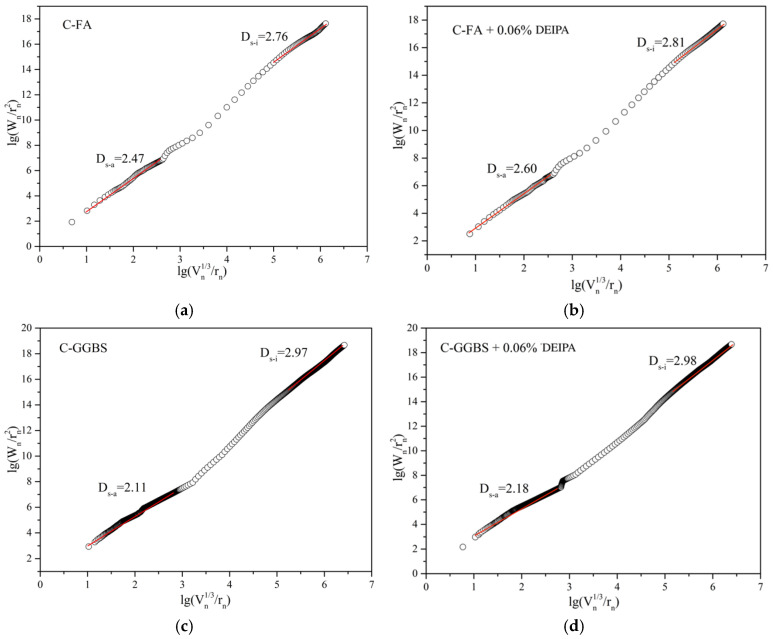
Logarithm plots of W_n_/r_n_^2^ versus V_n_^1/3^/r_n_ of C-FA/GGBS systems at 60 d, (**a**) C-FA, (**b**) C-FA with 0.06% DEIPA, (**c**) C-GGBS, (**d**) C-GGBS with 0.06% DEIPA.

**Table 1 materials-13-04103-t001:** Chemical compositions of raw materials.

Raw Materials		LOI.	CaO	MgO	SiO_2_	Al_2_O_3_	Fe_2_O_3_	SO_3_	K_2_O	Na_2_O
Cement	wt%	3.81	58.24	1.95	24.08	4.72	2.46	2.31	1.02	0.27
FA	wt%	5.97	4.12	0.50	48.33	31.69	4.14	1.37	1.34	0.37
GGBS	wt%	4.35	39.37	10.84	30.01	14.02	0.42	2.49	0.30	0.43

**Table 2 materials-13-04103-t002:** Mix proportions of samples (wt.%).

System	Cement	GGBS	FA	NaCl	DEIPA
C-GGBS	70	30	-	1.11	0
	70	30	-	1.11	0.03
	70	30	-	1.11	0.05
	70	30	-	1.11	0.10
C-FA	70	-	30	1.11	0
	70	-	30	1.11	0.03
	70	-	30	1.11	0.05
	70	-	30	1.11	0.10

**Table 3 materials-13-04103-t003:** Hydration heat within 7 d age.

Time (h)	Hydration Heat (J/g)					
	C-FA			C-GGBS		
	0.0%	0.06%DEIPA	0.10%DEIPA	0.0%	0.06%DEIPA	0.10%DEIPA
6	8.20	9.02	8.83	23.00	30.06	24.90
12	24.04	24.21	23.78	53.00	58.11	51.84
24	91.03	93.49	96.19	126.00	144.38	136.61
72	179.05	187.23	194.27	218.29	241.97	229.35
168	225.00	234.85	244.07	308.16	344.10	325.29

**Table 4 materials-13-04103-t004:** Loss weight of C-FA and C-GGBS samples cured for 7 d.

Temperature/°C	C-FA		C-GGBS	
	Blank	0.06% DEIPA	Blank	0.06% DEIPA
50–200 °C	8.49	9.25	10.55	11.71
280–390 °C	1.54	1.61	1.74	1.95
400–500 °C	2.64	2.71	2.40	2.42
CH content	10.85	11.14	9.87	9.95

**Table 5 materials-13-04103-t005:** Loss weight of C-FA and C-GGBS hydrated for 60 d.

Temperature/°C	C-FA		C-GGBS	
	Blank	0.06% DEIPA	Blank	0.06% DEIPA
50–200 °C	9.61	10.13	10.91	12.26
280–390 °C	1.98	2.13	1.82	1.95
400–500 °C	2.58	2.39	2.56	2.43
CH content	10.61	9.83	10.52	9.99

**Table 6 materials-13-04103-t006:** Porosity of C-FA and C-GGBS systems.

Age	C-FA (mL/g)		C-GGBS (mL/g)	
	Blank	0.06% DEIPA	Blank	0.06% DEIPA
7 d	0.1207	0.1336	0.1516	0.1510
60 d	0.0888	0.0902	0.1327	0.1133

**Table 7 materials-13-04103-t007:** Dissolution of Al in FA and GGBS suspension with and without DEIPA (mg/L).

Age	FA			GGBS		
	Blank	20 g/L DEIPA	Increase Ratio by DEIPA (%)	Blank	20 g/L DEIPA	Increase Ratio by DEIPA (%)
7 d	150	430	187	80	230	92
60 d	400	1400	367	150	700	100

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
