# Peer review of "Comparative Study on Chloride Binding Capacity of Cement-Fly Ash System and Cement-Ground Granulated Blast Furnace Slag System with Diethanol-Isopropanolamine"

_materials, 2020, doi:10.3390/ma13184103_

Round 1

Reviewer 1 Report

The authors present a paper on chloride binding capacity of cement-fly ash system and cement-ground granulated blast furnace slag system with diethanol isopropanolamine.

In general, the work is interesting, well organized, presents valid and interesting results and has potential for publication. However, some doubts arise that in my opinion should be clarified / corrected before the paper is considered for publication.

The issue of chloride binding capacity is indeed relevant for concrete subjected to a chloride contamination process, however, this fact is not clear in the abstract. It would be essential to explain in the abstract the need to develop this study. (however, in the introduction this fact is explained in more detail).

In the introduction, the current state of knowledge survey on this topic should be more detailed. It will be essential to present a current state of knowledge in more depth to justify the need for this study.

Following what was mentioned in the previous paragraph, it would be essential to present at the end of the introduction a justification for the need for this study, as well as its contribution to the current state of knowledge.

In point 2.3.1 the mix proportions are presented in a summary form, however, it would be very advisable that these values were properly organized in a table.

Will it be necessary to justify the two chosen test ages (7 and 60 d), why not at 90 d? or at 28d? or at 56 d?

without questioning the Chinese standard SL 352-2006, we have to consider that it may not be very well known and perhaps it would be interesting to detail a little more.

Regarding chapter 3 (Results and Discussion) it would be essential that a more detailed discussion of results was presented that included the benchmarking of the results.

In general, the conclusions could be more assertive, that is, which is better, which is worse, how they fit with reference values, among others.

Author Response

The authors present a paper on chloride binding capacity of cement-fly ash system and cement-ground granulated blast furnace slag system with diethanol isopropanolamine. In general, the work is interesting, well organized, presents valid and interesting results and has potential for publication. However, some doubts arise that in my opinion should be clarified / corrected before the paper is considered for publication.

Thanks for reviewing this manuscript. The manuscript was revised as your suggestion, and these contents were marked as red color.

  1. The issue of chloride binding capacity is indeed relevant for concrete subjected to a chloride contamination process, however, this fact is not clear in the abstract. It would be essential to explain in the abstract the need to develop this study. (however, in the introduction this fact is explained in more detail).

This was revised in text.

  1. In the introduction, the current state of knowledge survey on this topic should be more detailed. It will be essential to present a current state of knowledge in more depth to justify the need for this study.

This was revised in the introduction section.

  1. Following what was mentioned in the previous paragraph, it would be essential to present at the end of the introduction a justification for the need for this study, as well as its contribution to the current state of knowledge.
  • In point 2.3.1 the mix proportions are presented in a summary form; however, it would be very advisable that these values were properly organized in a table.

The table was added as your suggestion.

  • Will it be necessary to justify the two chosen test ages (7 and 60 d), why not at 90 d? or at 28d? or at 56 d?

This is a good question. Because of GGBS with higher activity at the early age, 7-d age was considered. At the age of 56 d or 60 d, pozzolanic reaction of FA would take place obviously, and therefore, the 60-d age was considered.

  • without questioning the Chinese standard SL 352-2006, we have to consider that it may not be very well known and perhaps it would be interesting to detail a little more.

This was revised in the text.

  • Regarding chapter 3 (Results and Discussion) it would be essential that a more detailed discussion of results was presented that included the benchmarking of the results.

This was revised in the text.

  • In general, the conclusions could be more assertive, that is, which is better, which is worse, how they fit with reference values, among others.

This was revised in the text.

Reviewer 2 Report

This is a good quality paper which investigates binding capacity of chloride in cement-fly ash and cement-ground granulated blast furnace slag systems with diethanol-isopropanolamine (DEIPA). It provides detailed results of study and concluded that DEIPA facilitates hydration and  promotes binding capacity of chloride.

Authors focus the importance of their study for construction materials although it is worth noting that the results obtained might be of practical use in nuclear waste immobilisation where cements play an important role see e.g. the monograph “Cementitious Materials for Nuclear Waste Immobilization” by R.O. Abdel Rahman, R.Z. Rahimov, N.R. Rahimova, M.I. Ojovan. Wiley, Chichester (2015).

The paper is recommended to publication practically as submitted apart from (a) requesting in the introductory part to emphasise that binding of chloride should be in insoluble compounds in highly-basic waters as there are no words on water  solubility of both FS and KS; and (b) giving data on device and method used or explaining where are taken from the particle size distributions of FA, GGBS, and cement shown in Figure 1.

Author Response

1. This is a good quality paper which investigates binding capacity of chloride in cement-fly ash and cement-ground granulated blast furnace slag systems with diethanol-isopropanolamine (DEIPA). It provides detailed results of study and concluded that DEIPA facilitates hydration and promotes binding capacity of chloride.

Thanks for reviewing this manuscript. The manuscript was revised as your suggestion, and these places were marked as red color. 

2. Authors focus the importance of their study for construction materials although it is worth noting that the results obtained might be of practical use in nuclear waste immobilisation where cements play an important role see e.g. the monograph “Cementitious Materials for Nuclear Waste Immobilization” by R.O. Abdel Rahman, R.Z. Rahimov, N.R. Rahimova, M.I. Ojovan. Wiley, Chichester (2015).

Several references were added.

3. The paper is recommended to publication practically as submitted apart from (a) requesting in the introductory part to emphasise that binding of chloride should be in insoluble compounds in highly-basic waters as there are no words on water solubility of both FS and KS;

FS and KS was the main components for chemical binding of chloride, which was widely reported in the literature. And therefore, it is unnecessary to discuss whether FS and KS were insoluble or not. Many thanks for your suggestion.

4. (b) giving data on device and method used or explaining where are taken from the particle size distributions of FA, GGBS, and cement shown in Figure 1.

This was revised in the text.

Reviewer 3 Report

Memorandum

Subject: Review, July 14, 2020

  1. of Materials

Title: Comparative study on chloride binding capacity of cement-fly ash system and cement-

           ground granulated blast furnace slag system with diethanol-isopropanolamine

 Huaqing Liu, Yan Zhang *, Jialong Liu, Zixia Feng and Sen Kong  

China Electric Power Research Institute, Beijing 100192, China

              * Correspondence: [email protected]

Comments:

  1. The authors should include a nomenclature to define parameters and abbreviations used throughout the paper.
  2. The abstract needs to be re-written. In its present format, it only delivers a set of announcements with no clear declaration of what is the content of the paper is. There is a miss connection in between the information presented. The subject should focus on what is being presented and what was found.
  3. The section “Test Method” does not offer much explanation on what the tests were, it merely states the theory. This needs to be clarified.
  4. Figures 7 and 8 should be labeled. Key points of interest must be labeled to indicate what is being observed.
  5. The authors in section 3, use subtitles such as “TG’, “RXD” and “SEM”, the full name must be used.
  6. Figures 10 and 11 can be reduced to 2 single comparisons plots illustrating the difference under each condition. Further, the response shown on these plots is not clearly explained. The authors state that the “could increase the 305 complexity of the pore structure of both C-FA and C-GGBS, and this benefited to hinder the transport 306 of chloride ions”.  This needs to be clarified.
  7. The authors should include a photo or a schematic of the specimen including dimensions.
  8. The authors used the word “promoted “to assess the effect of the DEIPA in this work. The term “Promoted’ does not offer a strong notion to what the effects are, I recommend using a substitute terminology.
  9. The paper needs major editorial workup. A lots of repetitive wordings  e. “in this study”, and grammatical issues exist.
  10. The paper is somewhat long, it may be beneficial if the authors focus on the work performed and the key findings instead of adding unnecessarily non relevant materials. A review to this option is recommended.
  11. The paper can use better organization, it is somewhat hard to follow.
  12. The conclusion can be improved by citing in bullet format the key finds and any issues that may have added to improving or affected the outcome of the results. In its present format, it only cites a set of not very conclusive statements.

Overall conclusion: the paper needs some considerable work, upon addressing the above issues, it can be ready for publication.

Author Response

Thanks for reviewing. the manuscript was revised as your suggestion, and these was marked as red color. 

  1. The authors should include a nomenclature to define parameters and abbreviations used throughout the paper.

Each abbreviation was defined as the first appearance. This was added in the text.

  1. The abstract needs to be re-written. In its present format, it only delivers a set of announcements with no clear declaration of what is the content of the paper is. There is a miss connection in between the information presented. The subject should focus on what is being presented and what was found.

This was revised in the text.

Steel bar corrosion caused by chloride was one of the main forms of concrete deterioration. Promotion of chloride binding capacity of cementitious materials would hinder the chloride transport to the surface of steel bar, thereby alleviating the corrosion and mitigating the deterioration. In this study, comparative study on binding capacity of chloride in cement-fly ash system (C-FA) and cement-ground granulated blast furnace slag system (C-GGBS) with diethanol-isopropanolamine (DEIPA) was investigated. Chloride ions was introduced by adding NaCl in paste, and the chloride binding capacity of the paste samples at 7 d and 60 d was examined. The hydration process was discussed via the testing of hydration heat and compressive strength. The hydrates in hardened paste was characterized by X-ray Diffractometry (XRD), Thermo Gravimetric Analysis (TGA), and Scanning Electron Microscope (SEM). Effect of DEIPA on dissolution of aluminate phase and compressive strength was discussed as well. These results showed that DEIPA could facilitate hydration of C-FA and C-GGBS system, and the promotion effect was higher in C-FA than that in C-GGBS. DEIPA also increased the binding capacity of chloride in C-FA and C-GGBS systems. One reason was the increased chemical binding, because DEIPA facilitated the dissolution of aluminate to benefit the formation of Friedel's salt. Other reasons were the increased physical binding and migration resistance. By contrast, DEIPA presented greater ability to increase chloride binding capacity in C-FA system, because DEIPA showed stronger ability to expedite the dissolution of aluminate of FA than that of GGBS, which benefited the formation of FS, thereby promoting the chemical binding. Such results would give deep insight into using DEIPA as an additive in cement-based materials. 

  1. The section “Test Method” does not offer much explanation on what the tests were, it merely states the theory. This needs to be clarified.

These were revised in the text.

  1. Figures 7 and 8 should be labeled. Key points of interest must be labeled to indicate what is being observed.

This was revised in the text.

  1. The authors in section 3, use subtitles such as “TG’, “RXD” and “SEM”, the full name must be used.

These were revised in the text.

  1. Figures 10 and 11 can be reduced to 2 single comparisons plots illustrating the difference under each condition. Further, the response shown on these plots is not clearly explained. The authors state that the “could increase the 305 complexity of the pore structure of both C-FA and C-GGBS, and this benefited to hinder the transport 306 of chloride ions”. This needs to be clarified.

If the curve in one figure, the curve would be covered and it seemed not very clearly. The text mentioned by reviewer was revised.

“The microporous structure could also be reflected by fractal dimension which was related to the complexity of pore structure.”

“This indicated that addition of DEIPA in both C-FA and C-GGBS systems could increase the complexity of pore structure.”

  1. The authors should include a photo or a schematic of the specimen including dimensions.

The dimensions of specimens were given in the text.

  1. The authors used the word “promoted “to assess the effect of the DEIPA in this work. The term “Promoted’ does not offer a strong notion to what the effects are, I recommend using a substitute terminology.

This was revised in the text.

  1. The paper needs major editorial workup. A lots of repetitive wordings “in this study”, and grammatical issues exist.

This was revised in the text.

  1. The paper is somewhat long, it may be beneficial if the authors focus on the work performed and the key findings instead of adding unnecessarily non relevant materials. A review to this option is recommended.

Thanks for your suggestion.

  1. The paper can use better organization, it is somewhat hard to follow.

Thanks for your suggestion. Some part was revised.

  1. The conclusion can be improved by citing in bullet format the key finds and any issues that may have added to improving or affected the outcome of the results. In its present format, it only cites a set of not very conclusive statements.

This was revised in the text.

Effect of DEIPA on chloride binding capacity of cement-fly ash system and cement-ground granulated blast furnace slag system was systematically discussed, and the conclusion was drawn as follow:

(1) DEIPA facilitated the hydration and increased the compressive strength of both C-FA and C-GGBS systems, and the main reason was that DEIPA facilitate the pozzolanic reaction of FA and GGBS.

(2) DEIPA increased the chloride binding capacity of both C-FA and C-GGBS systems. One reason was that DEIPA facilitated the dissolution of aluminate to hasten the formation of FS, with contribution to the chemical binding. Other reasons were that DEIPA increased the complex of pore structure which increased the migration resistance and also expedited the hydration of the system to produce more amount of C-S-H gel which benefited to the physical binding.

(3) By contrast, DEIPA presented greater ability to elevate the chloride binding capacity in C-FA system. The reason for this was that DEIPA showed stronger ability to expedite the dissolution of aluminate of FA than that of GGBS.

Round 2

Reviewer 1 Report

the authors responded adequately to the questions posed by the reviewers. In these conditions, I am of the opinion that the paper can be considered for publication.

Reviewer 3 Report

The paper reads much better, it can be released for publication.